# The Impact of the COVID-19 Pandemic on Vulnerable People Suffering from Depression: Two Studies on Adults in France

**DOI:** 10.3390/ijerph18063250

**Published:** 2021-03-21

**Authors:** Natalia Martinelli, Sandrine Gil, Johann Chevalère, Clément Belletier, Guillaume Dezecache, Pascal Huguet, Sylvie Droit-Volet

**Affiliations:** 1CNRS, LAPSCO, Université Clermont-Auvergne, F-6300 Clermont-Ferrand, France; natalia.martinelli@uca.fr (N.M.); johann.chevalere@hotmail.fr (J.C.); clement.belletier@uca.fr (C.B.); guillaume.dezecache@uca.fr (G.D.); pascal.huguet@uca.fr (P.H.); 2CNRS, CeRCA, Université de Poitiers, F-86000 Poitiers, France; sandrine.gil@univ-poitiers.fr

**Keywords:** COVID-19, depression, mental health, psychology, emotion, survey

## Abstract

This study investigated the difficulties experienced by people suffering from depression in coping with the stressful context of the COVID-19 pandemic and the lockdown. Two large samples of the French population were classified on the basis of their depressive symptoms and completed an online questionnaire on their emotions and their behaviors during the lockdown. Results showed that, compared to participants with no or mild mental health-related symptoms, participants with moderate to severe depressive symptoms suffered from greater psychological effects of the pandemic and the lockdown (fear, anxiety, sadness, sleep quality, loss of daily routine). However, health risk behaviors (smoking, drinking, non-compliance with lockdown and barrier gestures) and perceived vulnerability did not differ between the participant groups, although more severely depressed participants tended to be less respectful of health guidelines. In addition, the most heightened effects on the depressed participants were boredom and the feeling of social isolation, which was not compensated by the search for social affiliation. Supporting people with depression should be a public health priority because they suffer psychologically more than others from the pandemic and the lockdown.

## 1. Introduction

Depression has been declared a public health priority. According to the World Health Organization (WHO), depressive disorders are the leading cause of the global burden of disease and the associated economic burden [1]. A growing number of studies conducted worldwide suggest that the COVID-19 pandemic and its management—lockdown measures—have resulted in increased levels of depression in the general population [2,3,4]. However, they have not examined how people suffering from depression have coped with this pandemic and the lockdown. This vulnerable population may indeed require specific public health interventions for mental health. It is essential to maintain their wellbeing during the COVID-19 period in order to prevent the chronicity of depressive symptoms and their aggravation.

Studies that provide statistical data on the psychological effects of the COVID-19 pandemic indicate an increase in psychological distress (stress, fear, anxiety) [5,6,7]. These negative emotions are frequent in people exposed to an infectious disease [8]. A recent review of previous studies on nationally imposed quarantines for various pandemics (SARS, Ebola, H1N1 influenza) reminds us of their negative psychological consequences. Quarantine measures may even cause symptoms of post-traumatic stress when they are of long duration (>10 days) [9]. During the COVID-19 lockdown, people also reported feeling very bored and found that time slowed down [5,6,7,8,9,10]. Moreover, a significant proportion of people engaged in health risk behaviors, such as drinking alcohol [11,12].

It is well established that some of these psychological problems are habitual in depressed people. Depressed people are more anxious as anxiety is closely linked to depression [13]. They feel overwhelmed by sadness, by a loss of energy and decreased interest in activities. They suffer more from social isolation [14] and insomnia [15]. As a result, the COVID-19 pandemic and the lockdown measures may have greatly amplified depressed people’s initial difficulties. This might have made life during lockdown unbearable for them compared to those with little or no depression. They may also have been at risk of developing health risk behaviors such as the use of tobacco, alcohol and/or cannabis. In a recent study conducted in Australia, participants with moderate depressive symptoms reported suffering from sleep disorders and higher alcohol consumption [16]. It is also possible that depressed people might have been willing to take more risks by not closely following lockdown measures (e.g., leaving home more frequently) or by ignoring preventive measures (i.e., reduced observance of social physical distancing). However, this seems to depend on their level of fear or anxiety during the COVID-19 pandemic. Conversely, if their level of fear and anxiety is very high, they may also be more likely to comply with lockdown and preventive measures.

The aim of this study was to examine how people with moderate and severe depressive symptoms experienced the lockdown situation (e.g., feelings of isolation, emotion, sleep, substance consumption) and complied with the lockdown and preventive measures, as compared to people with no or fewer depressive symptoms. In France, the rules of lockdown were strict with only one hour of authorized outings (food shopping, individual physical exercise, pet needs, travel permits for professional activity that could not be carried out by teleworking) with a certificate of exit and police controls. However, the majority of workers kept their salary to stay at home during the lockdown. A survey was therefore conducted with French people (more than 2000 participants) in these lockdown conditions.

## 2. Study 1

### 2.1. Materials and Methods

#### 2.1.1. Participants

During the lockdown, a total of 1175 French participants (889 women and 286 men) completed an online questionnaire. There were 124 additional participants (9.55%) who started the survey, but did not complete the depression scale (BDI) used in our study. The participants gave their informed consent by agreeing to complete the questionnaire. This survey was conducted in accordance with the Declaration of Helsinki ethical principles and was approved by the Research Ethics Committee of the University Clermont Auvergne (IRB00011540-2020-31). Participants’ anonymity was guaranteed and they were allowed to stop answering the questionnaire at any time. 

#### 2.1.2. Procedure

The survey was created with the LimeSurvey software. It was distributed on online social networks during the French lockdown period from 1 to 29 April 2020. The survey consisted of a large set of questions and self-reported scales and took 40 minutes to complete. Here, we only report the results for the questions and scales of interest. We focused on reported emotion (boredom, anxiety, happiness, fear, anger, arousal), sleep quality, regularity of the rhythm of life and product consumption (alcohol, sleeping pills and anxiolytics, coffee/tea, cigarette and electronic cigarettes, stimulants, other substances) before and during the lockdown. The participants responded on a 7-point scale for most of the questions, and a 4-point scale for the consumption behaviors. There was also a series of questions with an 8-point response scale on self-reported compliance with the lockdown and respect for preventive measures (lockdown compliance and barrier gestures), perceived vulnerability (probability of being affected by the virus) and affiliation behaviors (i.e., contact with others via telephone, internet and social networks). For these last questions (compliance and affiliation), we used the average of the participants’ answers to the corresponding questions. The scales used in this study were the BDI (Beck Depression Inventory) for the assessment of depressive symptoms [17] and the 10-item short version of the UCLA loneliness scale for rating feelings of social isolation [18]. Their reliability was satisfactory (α = 0.82; α *=* 0.84, respectively).

#### 2.1.3. Statistical Analyses

We used IBM SPSS-25 (IBM Corp., Armonk, NY, USA) for our statistical analyses. We ran a series of ANOVAs with repeated measures with the depression group as between-subjects factor (depression) and the considered period (before vs. during the lockdown) as within-subject factor (lockdown). A Greenhouse–Geisser correction was applied when the sphericity assumption was violated. When a significant depression effect or depression x lockdown interaction was found, Student *t* tests were performed for two-way comparisons, with the *p*-values adjusted after the false discovery rates (FDR) correction (i.e., *q*-value) had been performed using the Benjamini–Hochberg approach [19]. The statistical analyses showed numerous significant results with *p*-values < 0.05, but with small effect sizes. We therefore considered the significant results only when the corresponding effect size was medium or large (Cohen’s *d* > 0.40 and partial eta-squared (η^2^_p_) > 0.06). 

### 2.2. Results

Figure 1 shows the percentage of participants according to their depressive symptoms (BDI scale); 50.2% of participants were non-depressive (N = 590), 26.7% (N = 314) presented mild depressive symptoms and 23.1% (271) showed moderate or severe depressive symptoms (18.6% (219) and 4.4% (52), respectively). As reported in Table 1, the groups did not differ in education level (*p* > 0.05) (M_no_ = 14.85, SD = 3.11; M_mild_ = 15.01, SD = 3.06; M_mod_*_-_*_sev_ = 14.62, SD = 2.63) or in their living conditions during lockdown, either in terms of living space (assessed in square meters) (M_no_ = 108.15, SD = 57.39; M_mild_ = 104.33, *SD* = 57.2; M_mod_*_-_*_sev_ = 97.82, SD = 60.53) or the number of people sharing the living space (including the participant) (M_no_ = 2.71, SD = 1.27; M_mild_ = 2.67, SD = 1.24; M_mod_*_-_*_sev_ = 2.79, SD = 1.30, all *p* > 0.05). However, the older the participants were, the less severe their depression tended to be (M_no_ = 45.68, SD = 15.67; M_mild_ = 40.31, SD = 15.81; M_mod_*_-_*_sev_ = 32.98, SD = 13.30; B = −0.099, β = −0.312, t = −11.67, *p* < 0.0001, 95% CI [−0.116; −0.082]). 

Our results (Table 1) indicate that the participants with depressive symptoms tended to report having complied with the lockdown rules and the preventive measures, just like the other participants (F(2, 1130) = 2.52, *p* = 0.11; F(2, 1114) = 14.32, *p* < 0.0001, η^2^_p_ = 0.025, respectively). Nevertheless, people with more severe signs of depression tended to be less compliant with preventive measures compared to the non-depressive group, Cohen’s d = 0.395, *q* < 0.05 (Table 2). The perception of vulnerability to COVID-19 was almost identical across groups, F(2, 1172) = 3.71, *p* = 0.025, η^2^_p_ = 0.006.

The major difference between the depression groups during the lockdown lay in a stronger feeling of social isolation (loneliness scale, F(2, 1116) = 156.33, *p* < 0.0001, η^2^_p_ = 0.22) (Figure 2). The feeling of social isolation increased with the severity of the depressive symptoms, B = 0.501, ES = 0.026, β = 0.503, t = 19.46, *p* < 0.0001, 95% CI [0.451; 0.552]. 

For all participants, affiliation needs increased significantly during the lockdown as compared to before, F(1, 1123) = 131.68, *p* < 0.0001, η^2^_p_ =.11. There was nevertheless little evidence that depressed participants sought more contact through social media than the other participants, F(2, 1123) = 10.74, *p* < 0.0001, η^2^_p_ =.019, especially during the lockdown period, as suggested by the non-significant interaction between depression and lockdown, F(2, 1123) = 0.22, *p* = 0.81.

Finally, our statistical analyses showed a large effect of depression on several emotional feelings (Figure 3); i.e., anxiety, F(2, 1172) = 158.76, *p* < 0.0001, η^2^_p_ = 0.21; boredom, F(2, 1172) = 97.92, *p* < 0.0001, η^2^_p_ = 0.14; low-arousal, F(2, 1172) = 76.49, *p* < 0.0001, η^2^_p_ = 0.115; and happiness, F(2, 1172) = 203.39, *p* < 0.0001, η^2^_p_ = 0.258 (Table 1). Depression also affected the emotions of fear and anger, but to a lesser extent (F(2, 1172) = 58.08, *p* < 0.0001, η^2^_p_ =.09; F(2, 1172) = 60.05, *p* < 0.0001, η^2^_p_ = 0.09, respectively). In addition, the feelings of boredom, F(1, 1172) = 388.78, *p* < 0.0001, η^2^_p_ = 0.249, fear F(1, 1172) = 411.16, *p* < 0.0001, η^2^_p_ = 0.26, and of sadness (lower scores on happiness scale), F(1, 1172) = 275.12, *p* < 0.0001, η^2^_p_ = 0.19, increased during lockdown (compared to before) for all participants (Figure 3). The other feelings, i.e., anxiety and anger, also increased, but only moderately, F(1, 1172) = 79.495, *p* < 0.0001, η^2^_p_ = 0.064, F(1, 1172) = 104.32, *p* < 0.0001, η^2^_p_ = 0.08. No difference was observed for low-arousal, F(1, 1172) = 1.25, *p* = 0.27. The effect size of the interaction between depression and lockdown (all η^2^_p_ < 0.04) suggests little difference between the depression groups in terms of the emotions felt during lockdown compared to before, except in the case of boredom, F(2, 1172) = 33.24, *p* < 0.0001, η^2^_p_ = 0.05. The increase in the boredom felt during compared to before lockdown (difference index) was greater in the participants with more severe depressive symptoms than in their counterparts, both those with mild depression (Cohen’s d = 0.329, *q* < 0.05) and those with no depressive symptoms (Cohen’s d = 0.597, *q* < 0.05), with the latter two groups of participants differing slightly in their levels of boredom (Cohen’s d = 0.25, *q* = 0.05). Indeed, the increase in boredom during lockdown was greater the higher the level of depression reported by the participants (B = 0.644, ES = 0.077, β = 0.237, t = 8.37, *p* < 0.0001, 95% CI [0.493; 0.795]. In short, the lockdown had the same emotional effects on all the participants, although it caused more boredom in those who were depressed. However, because the more depressed participants usually felt negative effects (fear, happiness, anxiety, anger) more strongly (compared with their less depressed counterparts), their reported emotions reached a critical level during the lockdown.

The participants also reported that lockdown disrupted the regularity of their life rhythm, F(1, 1172) = 188.83, *p* < 0.0001, η^2^_p_ = 0.14, and affected their sleep, albeit moderately, F(1, 1172) = 91.38, *p* < 0.0001, η^2^_p_ = 0.07. Furthermore, our results showed that depression itself disrupted the rhythm of life and the quality of sleep (F(2, 1172) = 49.29, *p* < 0.0001, η^2^_p_ = 0.08, F(2, 1172) = 54.41, *p* < 0.0001, η^2^_p_ = 0.09). Consequently, the lockdown increased problems related to daily rhythm and sleep in participants with depressive symptoms, although few differences were observed between the depression groups as a function of the periods considered, i.e., before and during lockdown (depression x lockdown interaction, F(2, 1172) = 21.68, *p* < 0.0001, η^2^_p_ = 0.036; F(2, 1172) = 20.48, *p* < 0.0001, η^2^_p_ = 0.034). Indeed, although all the comparisons between groups were significant, the only noteworthy differences were between the participants without and those with moderate to severe depression for the before/during lockdown difference index for both life rhythm (0.36 vs. 1.28, Cohen’s d = 0.48, *q* < 0.05) and sleep (0.09 vs. 0.78, Cohen’s d = 0.47, *q* < 0.05) (Table 2). It thus appears that quality of sleep and life rhythm was more disturbed in the depressed people during lockdown (Figure 4).

The reported consumption of sleeping pills and anxiolytics, F(1, 1172) = 7.17, *p* = 0.01, η^2^_p_ = 0.006, cigarettes, F(1, 1172) = 25.89, *p* < 0.0001, η^2^_p_ = 0.02, coffee/tea, F(1, 1172) = 50.83, *p* < 0.0001, η^2^_p_ = 0.042, stimulants, F(1, 1172) = 19.84, *p* = 0.0001, η^2^_p_ = 0.017, and other substances, F(1, 1171) = 29.98, *p* < 0.0001, η^2^_p_ = 0.025, varied little between the periods before and during the lockdown. The consumption of other substances such as cannabis even tended to decrease (.17 vs. 11). Individuals only reported drinking more alcohol during the lockdown, F(1, 1172) = 116.17, *p* < 0.0001, η^2^_p_ = 0.09. Furthermore, there was little or no difference in self-reported consumption across the groups, with depressed participants not drinking more alcohol than the other groups (depression, F(2, 1172) = 0.84, *p* = 0.43; depression x lockdown, F(2, 1172) = 2.57, *p* = 0.08). The only exception was found in the consumption of other substances such as cannabis (depression, F(2, 1171) = 19.35, *p* < 0.0001, η^2^_p_ = 0.09; depression x lockdown, F(2, 1172) = 3.99, *p* = 0.02, η^2^_p_ = 0.007), with a modest difference between the participants without and with moderate to severe signs of depression (Cohen’s d = 0.47, *q* < 0.05, Table 2).

## 3. Study 2

The aim of Study 2 was to replicate and extend the results of Study 1 with a new sample of participants. Indeed, the first sample, contacted via our social networks, was not representative of the population as a majority of women (76%) responded to the survey. Therefore, we recruited a new and more balanced sample by using a survey company (EasyPanel). This survey started three weeks after the first one on 24 April and ended on 28 April 2020.

### 3.1. Methods

The procedure and statistical analyses were similar to those used in Study 1. Unlike Study 1, however, Study 2 involved an equal proportion of men and women. The sample was composed of 1039 new participants: 526 women and 513 men. There were 156 missing values (13.05%) due to participants who did not complete the survey. All participants gave informed consent after reading the ethics form approved by the Research Ethics Committee of the University of Clermont-Auvergne, France (IRB00011540-2020-31).

### 3.2. Results

The BDI scores showed that 53% of participants did not show signs of depression (N = 551, 258 women, 293 men), against 23.9% who presented signs of mild depression (N = 247, 138 women, 109 men) and 23.2% with signs of moderate or severe depression (N = 241, 130 women, 111 men) (Figure 1). The depression groups did not differ in their education level (*p* > 0.05) (M_no_ = 13.26, SD = 2.95; M_mild_ = 12.91, SD = 2.76; M_mod-sev_ = 12.88, SD = 2.85), and the number of people living with them during lockdown (*p* > 0.05) (M_no_ = 2.64, SD = 1.21; M_mild_ = 2.64, SD = 1.21; M_mod-sev_ = 2.50, SD = 1.23) (Table 3). However, the living space (expressed in sq. meters) tended to be smaller in those presenting more severe depressive disorders (M_no_ = 102.10, *SD* = 48.55; M_mild_ = 95.83, *SD* = 46.12; M_mod-sev_ = 90.29, SD = 43.86; B = −0.013, ES = 0.004, β = −0.105, t = −3.41, *p* = 0.001, 95% CI [−0.02; −0.005]). However, only a small difference was observed between the no-depression and the moderate-severe depression group (Cohen’s d = 0.25, *q* < 0.05). The younger the participants were, the more severe their depressive symptoms were likely to be (B = −0.03, ES = 0.12, β = −0.079, t = −2.56, *p* = 0.01, 95% CI [−0.05; −0.007]). However, the age difference between the no-depression and the moderate-severe group also remained very small (Cohen’s d = 0.22, *q* = 0.05).

The results (Table 3) are entirely consistent with the findings of Study 1. There were few differences in compliance with lockdown and barrier gestures between groups, as indicated by the small effect size of our significant results (F(2, 1036) = 8.55, *p* < 0.0001, η^2^_p_ = 0.016; F(2, 1036) = 18.99, *p* = 0.0001, η^2^_p_ = 0.035, respectively). The depressive group did not report feeling more vulnerable to the virus than the other groups of participants, F(2, 1036) = 21.33, *p* < 0.0001, η^2^_p_ = 0.04. The major difference between groups was the feeling of social isolation during the lockdown, F(2, 1036) = 153.24, *p* < 0.0001, η^2^_p_ = 0.23 (Figure 2). The more depressed the participants were, the more socially isolated they felt (B = 0.556, ES = 0.029, β = 0.509, t = 19.05, *p* < 0.0001, 95% CI [0.498; 0.613]). This feeling of isolation did not change with participants’ sex (*p* > 0.05).

The affiliation behaviors changed with depression (M_no_ = 0.98, SD = 0.83; M_mild_ = 1.04; SD = 0.81, M_mod-sev_. = 1.32, SD = 1.24, F(2, 1036) = 11.05, *p* < 0.0001, η^2^_p_ = 0.02, and with lockdown (M_before_ = 1.11, SD = 1.01; M_lock_ = 1.04, SD = 0.94, F(1, 1036) = 14.88, *p* < 0.0001, η^2^_p_ = 0.014), but the interaction between depression and lockdown was not significant (*p* > 0.05). We also found a non-significant effect for the sex variable (*p* > 0.05). However, the effect size for these significant main factors (depression, lockdown) remained small, suggesting that out-of-home contacts were limited each day, but with more contacts for the participants with moderate to severe depression than for the other participants (Cohen’s d(M_no_, M_mod-sev_) = 0.35; d(M_mild_, M_mod-sev)_ = 0.27, both *q* < 0.05). 

With regard to the emotions felt during compared to before the lockdown, the participants felt more bored (Figure 3) and less happy (F(1, 1036) = 322.81, *p* < 0.0001, η^2^_p_ = 0.24; F(1, 1036) = 425.36, *p* < 0.0001, η^2^_p_ = 0.29). Their levels of fear, anxiety and anger also dramatically increased with the experience of lockdown (F(1, 1036) = 579.29, *p* < 0.0001, η^2^_p_ = 0.36; F(1, 1036) = 195.62, *p* < 0.0001, η^2^_p_ = 0.16; F(1, 1036) = 183.37, *p* < 0.0001, η^2^_p_ = 0.15). The level of arousal also decreased, albeit to a lesser extent, F(1, 1036) = 72.81, *p* < 0.0001, η^2^_p_ = 0.06. In addition, with the exception of anger, which was characterized by a small effect, F(2, 1036) = 54.05, *p* < 0.0001, η^2^_p_ = 0.09, all these negative emotions were heightened by depression (boredom, F(2, 1036) = 62.80, η^2^_p_ = 0.11; happiness, F(2, 1036) = 120.45, η^2^_p_ = 0.19; fear, F(2, 1036) = 55.122, η^2^_p_ = 0.10; anxiety, F(2, 1036) = 87.42, η^2^_p_ = 0.14; all *p* < 0.0001). Consequently, the level of these emotions was particularly high during lockdown among those suffering from depression. However, as in Study 1, the effect size of the interaction between depression and lockdown remained small, although significant, for each emotion (all *p* < 0.01 and η^2^_p_ < 0.03).

The lockdown also disrupted the regularity of the rhythm of life and of sleep in participants (F(1, 1036) = 185.26, *p* < 0.0001, η^2^_p_ = 0.15; F(1, 1036) = 122.39, *p* < 0.0001, η^2^_p_ = 0.11). In addition, the participants suffering from depression usually experienced more problems with daily rhythm and sleep (F(2, 1036) = 28.89, *p* < 0.0001, η^2^_p_ = 0.06; F(2, 1036) = 56.99, *p* < 0.0001, η^2^_p_ = 0.10). Indeed, the participants with the highest level of depression had a more irregular rhythm of life and slept less well than the other participants, both those with no depression (Cohen’s d = 0.58 and 0.78, both *q* < 0.05, Table 2) and those with mild depression (Cohen’s d = 0.38 and 0.38, both *q* < 0.05, Table 2). The mild depression sample also reported suffering from these problems (sleep and life rhythm) compared to the non-depressive participants. However, the difference only reached significance for sleep (Cohen’s d = 0.457, *q* < 0.05) but not for life rhythm (Cohen’s d = 0.21, *q* = 0.05, Table 2). Consequently, when compared to their initial status, rhythm and sleep problems during lockdown were particularly high in the depressed groups. However, this only resulted in a small effect size at the level of the depression x lockdown interaction (F(2, 1036) = 10.69, *p* < 0.0001, η^2^_p_ = 0.02; F(2, 1036) = 13.89, *p* = 0.001, η^2^_p_ = 0.03).

As in Study 1, participants reported little or no lockdown-related change in their consumption of sleeping pills and anxiolytics, F(1, 1036) = 8.68, *p* = 0.003, η^2^_p_ = 0.008, cigarettes, F(1, 1036) = 0.26, *p* = 0.61, coffee/tea, F(1, 1036) = 2.39, *p* = 0.12, stimulants, F(1, 1036) = 1.25, *p* = 0.26, and other substances, F(1, 1036) = 2.09, *p* = 0.15, as well as of alcohol, F(1, 1036) = 21.24, *p* = 0.0001, η^2^_p_ = 0.02. The depression x lockdown interaction was not significant for any of these consumed products (all *p* > 0.05). Only the consumption of sleeping pills and anxiolytics changed across the depression groups, F(2, 1036) = 35.76, *p* < 0.0001, η^2^_p_ = 0.07. As might be expected, their consumption increased with the severity of the depressive symptoms (B = 0.034, ES = 0.003, β = 0.291, t = 9.79, *p* < 0.0001, 95% CI [0.027; 0.04]).

## 4. Discussion

Our two studies show a prevalence of depression of about 23% of people with moderate and severe depressive symptoms, and of 27% with mild depressive symptoms. Data on the prevalence of depression in France are quite rare and vary between studies according to the criteria applied and the methods used to assess depression [20,21]. The prevalence rate is close to 8%–10%, a number that corresponds to the average global rate [22]. Therefore, in France, as in other countries, the number of depressive episodes experienced by participants increased during the lockdown period [2,3,4].

In our research we carried out two surveys, each on a large sample of participants, which provided entirely consistent results. These results revealed no or only a slight difference in demographic factors (age, education, living space) between the participants in the different depression groups. As several studies have suggested, demographic factors have little influence on the psychological effects related to a pandemic [8]. However, a study conducted in Turkey found that people living in urban areas are at higher risk of depression [23], potentially owing to differences in the quality of living conditions between the examined populations. In our research, the participants in the moderate or severe depression groups tended to be younger than those in the other groups. Depression scores are known to reach their lowest level at about 40–50 years of age [24], with higher economic levels in middle age being one of the factors that protect from depression. Our results also showed that a large majority of people abode by the lockdown and took preventive measures, even though those with more severe depressive symptoms tended to be less respectful of health guidelines. As a result, few participants, even those suffering from mental illness, were concerned about becoming infected with COVID-19. This finding was observed in an older study on the quarantine imposed during the H1N1 influenza pandemic in 2009 [25].

Provided responses to our survey were not influenced by social desirability, our results contradict the cliché that the French population is undisciplined and does not respect preventive rules. As in many other nations in the world, the French population appeared to be relatively afraid of the COVID-19 pandemic and stayed home. Our results on emotional factors indeed showed a significant increase in negative self-reported emotions in all participants during the lockdown period as compared to before (anxiety, fear, happiness, anger). This adds to the findings of studies indicating a high level of fear and stress felt in healthy populations during the COVID-19 pandemic, combined with a decrease in people’s level of happiness [5]. Nevertheless, our study indicated that the magnitude of the increase in these negative emotions was not greater for the participants with (vs. without) depression. However, because these negative emotions were already at a higher level (before lockdown) in the participants with moderate or severe depressive symptoms, they reached a particularly high emotional level with a mean of about 5 on a 6-point anxiety and fear scales. Self-reported levels of fear and anxiety in people with the most severe depression symptoms are particularly worrying.

Unlike in a survey conducted in Australia [16], this emotional context caused by COVID-19 did not result in higher self-reported health risk behaviors in our study. We noticed only a slight increase in reported alcohol consumption in Study 1, a finding not verified in Study 2. This is consistent with a recent Spanish study showing that alcohol consumption decreased as the lockdown continued [26]. The use of other substances such as cannabis even tended to decrease in our study in the first sample of participants. This can be explained by the fact that lockdown limits opportunities to buy and use prohibited substances, or that exposure to other family members further decreases consumption. Our statistical analyses only indicated that the participants with more severe depressive symptoms claimed to have a lower quality of sleep coupled with a more irregular daily routine during the lockdown than the other participants. This is consistent with studies showing more insomnia in depressed people [15]. In our study, however, sleeping problems in particular, increased during lockdown for all participants, including those with depression who already had a low quality of sleep.

In sum, the negative psychological effects of the lockdown in the context of COVID-19 were similar in all groups of participants, with or without depression, but these effects were nevertheless more severe in the participants with depression because of their vulnerability, i.e., their previous high level of psychological problems. Finally, our statistical analyses only found a significant interaction between depression and lockdown (before vs. after) factors for one emotional dimension, that is, boredom. The participants in the moderate/severe-depression group were more bored than those in the mild and the no-depression group. Their feeling of boredom was exacerbated during the lockdown as compared to the other groups. Indeed, their boredom scores were twice as high as those of non-depressed people. This was likely related to the feeling of social isolation. Indeed, in our research, the feeling of social isolation was very high in the participants with moderate or severe depressive symptoms compared to the others. In addition, our research suggests that these participants did not compensate for this sense of social isolation by increasing telephone or social network contacts. Affiliation behaviors increased during compared to before the lockdown, but not by much more in the depressed participants. This is likely due to the fact that individuals with depression usually have fewer social contacts [14]. As noted in recent COVID studies, during the lockdown, individuals had little physical activity and spent a lot of time alone in front a screen, which contributed to poor mental health [27,28].

One limitation of the present work was the classification of participants into different depression groups based of their scores on a self-reported scale (BDI), and not on advanced clinical diagnosis. Our results should therefore be taken with caution. They may reflect mental health problems rather than actual psychiatric illnesses. In addition, a depressive episode during the lockdown may not result in psychiatric illness. Therefore, it would be important to contact people to monitor their mental health problems. In addition, although participants described their behaviors prior to the lockdown, we do not have their depression score prior to this period (baseline). Therefore, among the participants tested there are likely people who have been depressed for a long time and people who have recently been depressed (contextual). In all cases, the pandemic and the lockdown caused intense psychological distress that can lead to psychiatric illness, especially if other periods of lockdown are imposed in response to successive waves of the pandemic, with an accumulation of stressful events (job loss, financial difficulties, illness, etc.).

## 5. Conclusions

Our studies using a questionnaire completed by two large samples of French participants confirmed the numerous negative psychological effects of the COVID-19 pandemic and of lockdown on the general population, and more particularly on depressed people. Among the most exacerbated negative effects observed in the depressed groups were boredom and the feeling of social isolation during lockdown. Supporting depressed people during the COVID-19 pandemic period should therefore be a public health priority because they suffer psychologically more than others from the pandemic and its management by health authorities.

## Figures and Tables

**Figure 1 ijerph-18-03250-f001:**
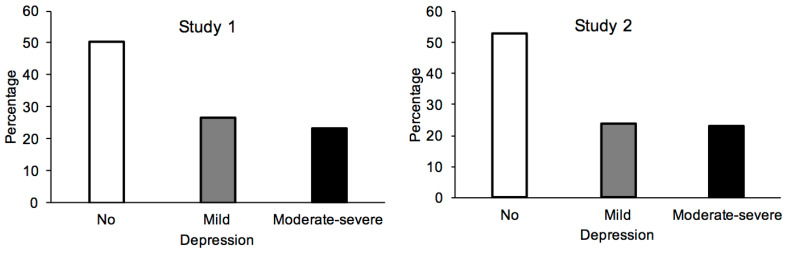
Severity of the depressive symptoms in Study 1 and Study 2.

**Figure 2 ijerph-18-03250-f002:**
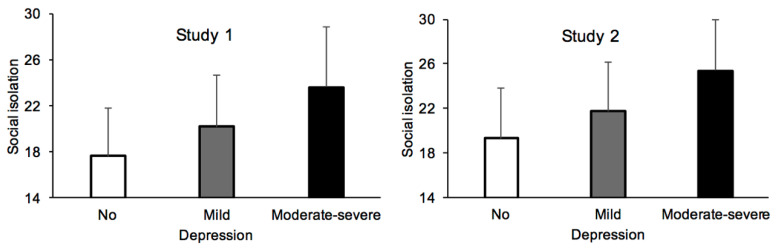
Feeling of social isolation as a function of severity of depressive symptoms in Study 1 and Study 2.

**Figure 3 ijerph-18-03250-f003:**
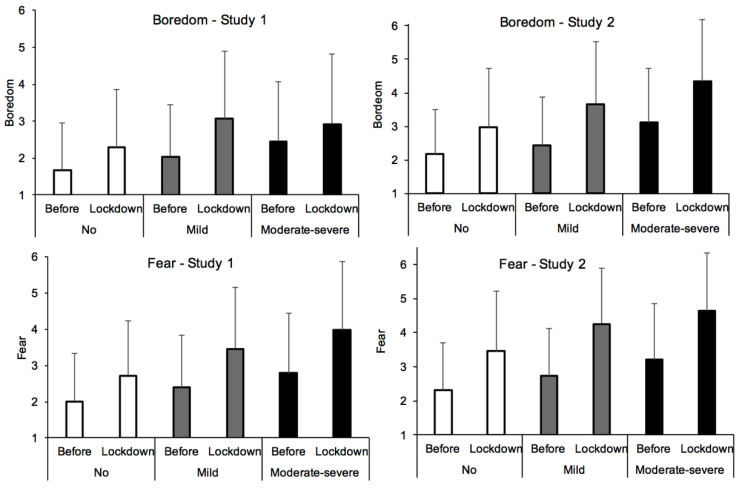
Feelings of boredom and of fear as a function of severity of depressive symptoms in Study 1 and Study 2.

**Figure 4 ijerph-18-03250-f004:**
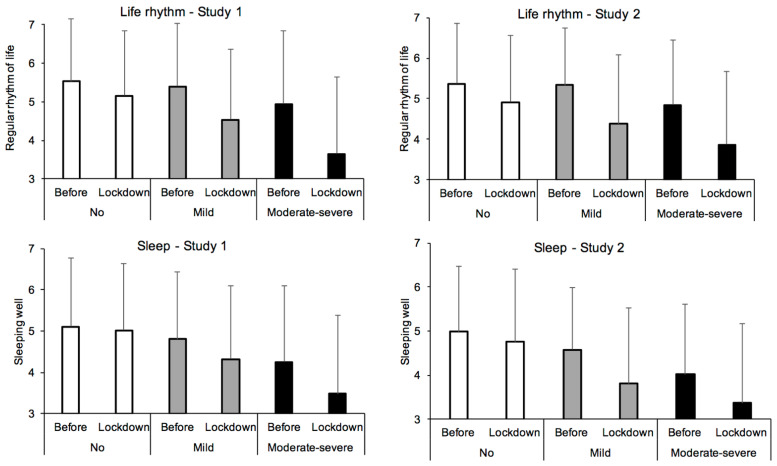
Quality of sleep and of life rhythm as a function of severity of depressive symptoms in Study 1 and Study 2.

**Table 1 ijerph-18-03250-t001:** Mean (SD) of the participants’ answers to the survey questions used in Study 1 according to depressive symptoms: no, mild and moderate-severe.

	No Depression				Mild Depression				Moderate-Severe Depression			
	Before		Lockdown		Before		Lockdown		Before		Lockdown	
	M	SD	M	SD	M	SD	M	SD	M	SD	M	SD
Age			45.68	15.67			40.31	15.81			32.98	13.30
Education			14.85	3.11			15.01	3.06			14.62	2.63
poeple home			2.71	1.27			2.67	1.24			2.79	1.30
Living space (m^2^)			108.15	57.39			104.33	57.2			97.82	60.53
Affiliation	0.88	0.62	1.08	0.68	0.92	0.57	1.11	0.596	1.07	0.74	1.30	0.80
Compliance lockdown			5.01	0.58			4.93	0.56			5.004	0.59
Compliance gestures			4.49	0.89			4.36	0.86			4.12	1.03
Vulnerability perception			1.61	0.96			1.68	0.99			1.81	1.09
Social isolation			17.67	4.19			20.21	4.49			23.64	5.21
Boredom	1.68	1.28	2.29	1.56	2.04	1.41	3.07	1.81	2.45	1.61	4.14	2.03
Anxiety	2.56	1.60	2.76	1.55	3.35	1.71	3.71	1.68	4.25	1.79	4.90	1.77
Happiness	5.49	1.13	5.14	1.18	5.01	1.20	4.40	1.22	4.23	1.56	4.50	1.46
Fear	2.01	1.32	2.72	1.51	2.39	1.46	3.46	1.70	2.79	1.66	3.98	1.89
Anger	2.31	1.56	2.48	1.66	2.80	1.74	3.22	1.80	3.14	1.70	4.01	1.85
Low-Arousal	4.13	1.54	4.44	1.48	3.84	1.57	3.90	1.59	3.22	1.55	3.03	1.54
Sleep	5.11	1.65	5.02	1.61	4.80	1.64	4.32	1.78	4.26	1.84	3.48	1.90
Life rhythm	5.53	1.62	5.16	1.69	5.40	1.64	4.54	1.82	4.94	1.90	3.65	2.00
Acohol	1.11	0.75	0.96	0.79	1.13	0.78	0.93	0.80	1.23	0.85	0.97	0.96
sleeping pills-anxiolytics	0.08	0.30	0.06	0.24	0.12	0.32	0.09	0.30	0.26	0.51	0.25	0.56
Coffee/tea	1.41	0.63	1.30	0.67	1.40	0.71	1.34	0.72	1.30	0.70	1.20	0.80
Cigarette	0.26	0.54	0.25	0.53	0.31	0.57	0.26	0.56	0.47	0.68	0.29	0.58
Stimulant	0.06	0.32	0.03	0.20	0.03	0.17	0.01	0.10	0.13	0.41	0.06	0.31
Other substances	0.07	0.33	0.04	0.25	0.14	0.48	0.10	0.43	0.30	0.67	0.20	0.64

**Table 2 ijerph-18-03250-t002:** Statistical results of comparisons between two groups (moderate-severe vs. mild vs. no-depression).

Group Comparison	Dependent Variable	Cohen’s d	ddl	*t*	*p*	Rank	*q ^a^*
*Study 1*							
Mod-Sev vs. No	Boredom	0.60	859	8.13	<0.001	1	=0.050
Mod-Sev vs. No	Life rhythm	0.48	859	6.56	<0.001	2	=0.043
Mod-Sev vs. No	Sleep	0.47	859	6.37	<0.001	3	=0.036
Mod-Sev vs. No	Other substances	0.47	859	6.37	<0.001	4	=0.029
Mod-Sev vs. No	Compliant gestures	0.40	814	5.24	<0.001	5	=0.021
Mod-Sev vs. Mild	Boredom	0.33	583	3.97	<0.001	6	=0.014
Mild vs. No	Boredom	0.25	902	3.59	<0.001	7	=0.007
*Study 2*							
Mod-Sev vs. No	Sleep	0.78	790	10.14	<0.001	1	=0.005
Mod-Sev vs. No	Life rhythm	0.58	790	7.46	<0.001	2	=0.01
Mild vs. No	Sleep	0.457	796	5.97	<0.001	3	=0.15
Mod-Sev vs. No	Affiliation	0.35	790	4.49	<0.001	4	=0.02
Mod-Sev vs. Mild	Life rhythm	0.38	486	4.21	<0.001	5	=0.025
Mod-Sev vs. Mild	Sleep	0.34	486	3.77	<0.001	6	=0.03
Mod-Sev vs. No	Living space	0.25	789	3.24	=0.001	7	=0.035
Mod-Sev vs. Mild	Affiliation	0.27	486	2.94	=0.003	8	=0.04
Mod-Sev vs. No	Age	0.22	790	2.84	=0.004	9	=0.045
Mild vs. No	Life rhythm	0.21	796	2.79	=0.01	10	=0.05

^a^*q*–values = FDR ajusted *p*-values using Benjamini & Hochberg’s approach.

**Table 3 ijerph-18-03250-t003:** Mean (SD) of the participants’ answers to the survey questions used in Study 2 according to depressive symptoms: no, mild and moderate-severe.

	No Depression				Mild Depression				Moderate-Severe Depression			
	Before		Lockdown		Before		Lockdown		Before		Lockdown	
	M	SD	M	SD	M	SD	M	SD	M	SD	M	SD
Age			46.62	15.03			46.65	15.36			43.38	13.92
Education			13.26	2.95			12.91	2.76			12.88	2.85
poeple home			2.64	1.21			2.64	1.21			2.50	1.23
Living space (m^2^)			102.10	48.55			95.83	46.12			90.29	43.86
Affiliation	1.00	0.88	0.96	0.83	1.08	0.89	1.00	0.81	1.35	1.30	1.29	1.25
Compliance lockdown			4.49	0.89			4.36	0.86			4.12	1.03
Compliance gestures			4.51	0.94			4.47	0.97			4.05	1.07
Vulnerability perception			1.43	0.97			1.51	1.00			1.95	1.23
Social isolation			19.32	4.54			21.73	4.44			25.40	4.55
Boredom	2.17	1.32	2.97	1.75	2.43	1.44	3.66	1.87	3.11	1.61	4.34	1.84
Anxiety	2.80	1.49	3.36	1.66	3.43	1.48	4.34	1.60	4.00	1.60	4.78	1.60
Happiness	5.30	1.12	4.67	1.31	4.86	1.09	3.85	1.29	4.28	1.41	3.15	1.48
Fear	2.31	1.38	3.46	1.75	2.72	1.40	4.26	1.61	3.21	1.64	4.63	1.71
Anger	2.45	1.39	3.13	1.78	2.90	1.43	3.74	1.76	3.39	1.58	4.30	1.75
Low-Arousal	4.54	1.35	4.24	1.45	4.09	1.36	3.48	1.43	3.69	1.44	3.20	1.59
Sleep	4.98	1.60	4.75	1.64	4.57	1.61	3.81	1.67	4.02	1.77	3.37	1.73
Life rhythm	5.37	1.49	4.91	1.65	5.34	1.41	4.38	1.71	4.85	1.59	3.87	1.81
Acohol	1.02	0.74	0.93	0.80	1.09	0.75	0.97	0.87	1.05	0.80	1.00	0.88
sleeping pills-anxiolytics	0.14	0.47	0.16	0.51	0.26	0.66	0.28	0.72	0.53	0.87	0.60	0.97
Coffee/tea	1.84	0.87	1.76	0.89	1.78	0.79	1.79	0.86	1.85	0.94	1.84	1.00
Cigarette	0.47	0.90	0.47	0.92	0.45	0.90	0.45	0.92	0.80	1.07	0.79	1.13
Stimulant	0.09	0.37	0.09	0.40	0.13	0.42	0.11	0.41	0.30	0.67	0.28	0.69
Other substances	0.07	0.35	0.06	0.35	0.07	0.32	0.09	0.41	0.19	0.55	0.22	0.65

## Data Availability

The data presented in these two studies are available on request from N.M. and S.D.-V. The data are not publicly available now because they are part of a larger project still in progress.

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
