# Peer review of "The Impact of the COVID-19 Pandemic on Vulnerable People Suffering from Depression: Two Studies on Adults in France"

_ijerph, 2021, doi:10.3390/ijerph18063250_

Round 1

Reviewer 1 Report

The manuscript entitled “The impact of the COVID-19 pandemic on vulnerable people suffering from depression: two studies on adults in France” by Martinelli et al. reports the results of surveys carried out on the French population investigating the effects of the lockdown on self-reported mild or severe depression versus a non-depressed group. High depression prevalence was discovered with increased negative emotions. Lockdown measures were generally followed by all groups; depressed patients reported lower respect of health measures, although risky behaviours did not differ among groups.

The findings are interesting and the manuscript is well written. However, the description of two separate studies is confusing, especially with methods and results displayed twice. Therefore, as it seems that both studies were performed in the same timeframe with the same methods on the same French population, I am not clear about which is the difference between them and what prevents the authors from merging the results of both studies. This approach would also allow gaining more power in the statistical analysis. Thus, my suggestion is to analyse all data together by introducing the study number as an additional factor to be included in the statistical analysis. If this is not possible because the surveys were carried out at different periods or with different methods, this should be clearly explained in an experimental design section and single Methods and Results sections including both studies should be reported.

A brief explanation better describing the lockdown measures, i.e. describing what was allowed and what was forbidden in the indicated periods of time in the country would make comparisons easier with studies carried out in other countries or with future studies.

References are required at line 29.

Author Response

Reviewer 1

The findings are interesting and the manuscript is well written.

--> Many thanks.

"However, the description of two separate studies is confusing, especially with methods and results displayed twice. Therefore, as it seems that both studies were performed in the same timeframe with the same methods on the same French population, I am not clear about which is the difference between them and what prevents the authors from merging the results of both studies. This approach would also allow gaining more power in the statistical analysis. Thus, my suggestion is to analyse all data together by introducing the study number as an additional factor to be included in the statistical analysis. If this is not possible because the surveys were carried out at different periods or with different methods, this should be clearly explained in an experimental design section and single Methods and Results sections including both studies should be reported.

à The text now includes the rationale for using two different samples.

A brief explanation better describing the lockdown measures, i.e. describing what was allowed and what was forbidden in the indicated periods of time in the country would make comparisons easier with studies carried out in other countries or with future studies.

--> The lockdown measures used in France were described at the end of the introduction

References are required at line 29.

--> Ok !

Many thanks for your reviewing work.

Reviewer 2 Report

This is an interesting manuscript dealing with a hot topic. However, several points need to be addressed.

Abstract

-It is necessary to provide information regarding sample characteristics, methods (statistics), and results.

Methods

-Something needs to be said about missing values.

-Provide information about the sampling method (was it at convenience?)

-You state that regression analyses were performed, but no coefficient or ORs was provided in the results or tables.

More details, find the attachment.

Author Response

Reviewer 2

This is an interesting manuscript dealing with a hot topic. However, several points need to be addressed.

--> Many thanks

Abstract

-It is necessary to provide information regarding sample characteristics, methods (statistics), and results.

--> It is not mandatory. It is also too long to give this information in the abstract composed of a reduced number of words and 2 studies.

Introduction

-Overall, it is quite redundant. Avoid writing the same or similar information twice.

--> As you suggested, we have tried to reduce redundancy.

-Lines 27-29. Provide a reference for such a statement regarding WHO.

--> We added this reference.

-Lines 31-32. The sentence “COVID-19 therefore has serious effects on the major mental health problem of depression.” It's redundant since this has been stated before.

--> This sentence has been removed.

-Lines 40-41. It is a bit contradictory to speak about limited evidence after having said that there is growing evidence about the effect of Covid-19 lockdown on depression. I would speak about lack of high-quality evidence instead or I would state something more specific.

--> Ok! We solved this contradiction.

-Information about the surveys should be moved to the Methods section.

-> I agree. This is too redundant.

Methods

-Provide information about the sampling method (was it at convenience?)

--> Ok !

-As commented above, there is information in the Methods section that corresponds to the Results section. For example, the number of participants presenting symptoms of depression is not suitable for this section (Studies 1 and 2).

--> This has been modified. Many thanks.

-Something needs to be said about missing values. At least, the percentage of those, and how did you cope with it (Studies 1 and 2).

--> We agree. We added this data for Study 1 and Study 2.

-Provide information about the sampling method (was it at convenience?) If it was, this must be stated in the limitations since your sample is not representative of the study population (Studies 1 and 2) .

--> This has been added in Study 2 before the method.

-You state that regression analyses were performed, but no coefficient or ORs was provided in the results or tables. Also, which statistical analyses were conducted for each study?

--> There are only one or two regression analyses. The coefficients are now set.

Discussion

-It is well conducted. However, you might want to highlight the notion that higher levels of watching TV and lower levels of physical activity during the COVID-19 confinement, might have also contributed to worsen mental health.

--> Many thanks. The two interesting references are not cited in our discussion.

-Revise typo of Line 483 “RiskBehaviors”

--> Thank you for all.

Reviewer 3 Report

Interesting paper about the difficulties experienced by people with depression in dealing with the stressful context of the COVID-19 pandemic and the lockdown. The results showed that participants with moderate to severe depressive symptoms experienced greater psychological effects of the pandemic and the blockade (fear, anxiety, sadness, quality of sleep, loss of daily routine). Supporting people with depression should be a public health priority because they suffer more psychologically than others from the pandemic and the lockdown, being relevant the publication of this article.

Author Response

Thank you for this clear summary and for approving our work.

Round 2

Reviewer 1 Report

The revised manuscript is suitable for publication.